# Fostering Offshore Wind Integration in Europe through Grid Connection Impact Assessment

Nuno Amaro *[ID], Aleksandr Egorov and Gonçalo Glória [ID]

R&D NESTER—Centro de Investigação em Energia REN State Grid, 2835-659 Sacavém, Portugal;
aleksandr.egorov@rdnester.com (A.E.); goncalo.gloria@rdnester.com (G.G.)
* Correspondence: nuno.amaro@rdnester.com

**Abstract:** Floating offshore wind energy is one of the solutions that can foster the ongoing climate transition in Europe. The ARCWIND project aims to contribute to this topic by considering multiple research activities designed to contribute to the development of multiple floating technologies, identifying high-potential deployment areas while considering their economic viability and the impact that these would have in existing power systems. Regarding the latter activity, a two-step methodology was implemented to first calculate the nodal capacity that existing electricity networks have to absorb energy from these potential new wind farms and secondly to assess the impact at the point of connection. This assessment is performed by identifying grid reinforcement needs, verifying the impact on short circuit current levels and measuring the impact on the existing energy mix at countrywide level. This article includes the description of this methodology as well as its application to six different use cases covering five European countries: Portugal, Spain, France, the United Kingdom, and Ireland. Results obtained seem to indicate that, in most cases, the current power systems have enough capacity for the possible connection of new floating offshore wind farms without major reinforcement needs and that these wind farms can have a major contribution to the countries' energy mix and to the achievement of established climate targets.

**Keywords:** nodal capacity; offshore wind; RES integration; ARCWIND

## 1. Introduction

Renewable energy sources are one of main solutions for decreasing carbon emissions in power systems; thus, contributing to the ongoing energy transition in Europe and other regions of the world [1]. A growing penetration of renewable energy sources and the technological developments leading to the advent of smart grids is directly correlated with an increasing number of grid connection requests, from the side of energy producers. This represents the distributed nature of renewable energy sources, when compared to fossil fuel energy generation units. One example of this positive correlation is Portugal, where the number of renewable energy generation units, also including Distributed Energy Resources (DER), have been increasing gradually. Renewable energy has already been used to supply around 50% of electricity consumption consecutively over recent years and the number of new connection requests and new connection projects from generators have been also increasing accordingly [2].

This increase in grid connection requests from generation units represents different technological challenges that need to be considered, particularly in order to safely integrate DER at a large scale [3]. From a grid planning perspective, one of these challenges relates to the requirement that System Operators (SOs) must be able to determine the capacity that each grid node has to absorb new energy sources, defined as nodal capacity, both in normal and under contingency conditions. This process allows SOs to verify which are the grid areas more prone to congestion issues and helps with the decision-making process related to asset management, investment, and grid expansion needs. In some countries, regulatory

authorities impose that SO's need to calculate and publish the nodal capacity of the different nodes in the power system, in the current and considered expansion scenarios in their grid development plans. In Portugal, REN, the Portuguese Transmission System Operator (TSO) calculates these capacities using a zonal approach (joint capacity of different nodes in the same zone), and publishes them in the Portuguese Transmission System network development plan [4].

However, the definition and calculation of nodal capacities does not have a univocal solution. SOs normally use in-house developed tools whose methodologies are not published, and different scientific groups deal with this process using different approaches as well. Such methods include the utilization of Optimal Power Flow (OPF) tools [5,6] and different meta-heuristics such as Genetic Algorithms and Particle Swarm Optimization techniques [7–10]. However, most of these works are related to the calculation of optimal locations for deployment of DER and not as a systemic analysis for the calculation of nodal capacities of grid nodes.

In previous research [11,12] we developed a Nodal Capacity Allocation tool, allowing a systemic vision on this topic. This methodology calculates the nodal capacity values of multiple user-defined grid nodes. Depending on the objective of the study and grid intrinsic characteristics, nodal capacities of the candidate nodes can be calculated using different approaches. The tool uses genetic algorithms to maximize the total value of power absorbed at grid level. Obtained results always represents valid technical solutions in terms of AC power flow and voltage profile (user-defined voltage limits), allowing it to comply with existing planning and operational criteria.

Wind is currently one of the largest energy resources used in Europe. In 2020, wind accounted for 16% of the electricity consumed in the EU27+UK area, with a total installed capacity of 220 GW [13]. From these, 194 GW are onshore solutions but offshore wind already represents an important share of deployed installations and during 2020 it represented a total of 20% of new installations commissioned [13]. In the offshore wind field, fixed installations still represent the vast majority of deployed solutions, due to the technical advancements when compared to floating wind solutions. Nonetheless, due to the nature of their coastal areas (mainly depth), different regions and countries need to focus on floating wind solutions, as is the case of most Atlantic areas in Europe. Multiple floating wind solutions are already installed or currently being tested in a commercial stage, as is the case of WindFloat Atlantic, with a total installed capacity of 25 MW over the Portuguese Coast [14]. In this context, the project ARCWIND aims to contribute to the development of floating wind solutions in multiple European Atlantic countries.

One of the research activities included in ARCWIND relates to the need to perform grid integration studies for potential floating wind installations in selected areas of interest. In this research activity, a two-step methodology was put into place. This methodology considers the calculation of the nodal capacity that power systems have to receive new connection requests, particularly from offshore wind solutions, and uses different metrics to perform a grid impact assessment study by evaluating possible grid expansion needs and the impact in terms of contribution to the existing energetic mix in each area under study.

In this paper, we present the outcomes of this activity, reporting the nodal capacities and grid impact assessment performed in six different areas included in five European countries: Portugal, Spain, France, Ireland, and Scotland. A previous study including the calculation of the nodal capacity of selected candidates nodes in the Portuguese Transmission Network was already published in [15] and results herein reported for Portugal only include the grid assessment study outcomes. For the other countries, the results of both are included. This work aims provide a contribution to decision makers to foster the development of the considered power systems and floating wind solutions, assessing how these will impact the power system developments and also its contribution to the achievement of current climate targets set at the European level.

**2. ARCWIND Project**

ARCWIND is an R&D project funded by the European Regional Programme INTER-REG Atlantic Area. It aims to contribute to the development of floating offshore wind technologies, focused on its application in European Atlantic areas in Portugal, Spain, France, the United Kingdom, and Ireland [16].

The project is composed of a set of multi-disciplinary activities, aimed at creating sound contributions to the development of these technologies. These activities include: (i) a multi-criteria methodology for wind assessment and site planning decision-making processes; (ii) laboratory size prototype development and testing of three different technologies of floating wind platforms; (iii) array design, installation logistics, and maintenance planning studies; (iv) electric grid integration related studies; (v) cost estimation and economic viability studies.

Considering the current development of floating offshore wind solutions and the potential impact that these might have in power systems, in the medium and long term, it is necessary to perform different studies related to the integration of these new sources in existing power systems. Under the scope of ARCWIND, a two-step methodology was considered, aiming at reaching two complementary goals: the calculation of the nodal capacities of candidate nodes in existing power systems and the grid analysis of future scenarios considering floating offshore wind farms in the project's selected areas.

The application of a multi-criteria methodology for wind assessment and site planning allowed to identify different areas of interest in the five countries considered in ARCWIND [17,18]. Considering 10 MW wind turbines, the maximum theoretical installed capacity was calculated for each one for these areas. These values, presented in Table 1, and together with the location of the areas are the only required input data for the grid impact assessment studies performed within the scope of the project.

**Table 1.** Identified areas of high potential and their maximum installed capacity.

| Country | Area Name | Maximum Installed Capacity (MW) |
|---|---|---|
| Portugal | Figueira da Foz | 700 |
| Spain | Ribadeo | 880 |
| Spain | Gran Canaria | 120 |
| France | Lannion | 470 |
| Ireland | Galway Bay | 98 |
| Scotland | Orkney | 154 |

**3. Grid Connection Impact Assessment Methodology**

In order to evaluate the impact that the connection of the potential wind farms would have on the existing power systems, a two-step methodology was implemented. These two steps are complementary and aim to evaluate the impact at local and national levels.

The nodal capacity calculation process aims at identifying the best possible grid node to perform the connection of the offshore wind farm, starting from a set of pre-established candidates.

After the best candidate grid node is identified and selected, the second step assumes that a wind farm with the previously calculated installed capacity is connected to this node and further impact assessment studies are performed. This methodology is applied to all studied countries and high potential areas.

*3.1. Nodal Capacity Calculation*

Nodal capacity can be defined as the capacity that each grid node has to receive new power sources in safe operational conditions. The nodal capacities were calculated by applying the methodology described in [11,12]. The methodology has two main requisites:

- the existence of a model of the power system under analysis, which allows the execution of steady state simulations (power flow);

- Manual selection of candidate nodes, which are the network buses (or substations) where the capacity will be calculated.

Nodal capacities are calculated by connecting new generation units to the candidate nodes and increasing generation levels until overloads are obtained in the network. This represents an optimization problem that is solved using genetic algorithms with a target function of increasing both the local and system level nodal capacities, through the evaluation of the fitness function of individual solutions in the population.

The selection of candidate nodes is highly correlated with the goal of the study to be performed. In this particular case, because the goal is to study the possible connection of new offshore wind farms (based on floating technologies), the following criteria was applied to all areas under study, in order to select candidate nodes:

- Geographic location of the substation;
- Adequacy of voltage level of the substation to the wind farm projected installed capacity;
- Exclusion of client substations (e.g., industrial facilities or generation unit substations);
- Exclusion of substations that can generate high congestion (e.g., if there is a single line connecting the substation to the remaining power system).

*3.2. Connection Impact Assessment*

The second step of the implemented methodology consists of a verification of the impact on the point of connection. This analysis is performed as follows:

- Selection of the best candidate node. This selection is performed taking into consideration the calculated nodal capacities for the different candidate nodes, together with the criteria for selection of the candidate nodes themselves. Therefore, for example, if two candidate nodes provide enough capacity to receive energy generated in the offshore wind farm, the one closer to the wind farm projected location is selected as the best candidate.
- Identification of grid expansion needs. In cases where the best candidate (the one with a higher nodal capacity) does not have enough capacity to absorb the wind farm energy in safe operational conditions (without voltage issues or line congestion), the necessary grid expansion needs will be indicated.
- Short circuit calculation and impact on protection systems. Three-phase short circuit currents are calculated for the selected grid node and neighboring ones in order to assess the impact of the connection of the wind farm on the increase in short circuit current levels. These are of upmost importance as protection systems (including circuit breakers) are designed based on short circuit current levels and its increase might result in the need to refurbish part of these systems. If deemed necessary, these refurbishment needs are highlighted.
- Impact on the energy mix of the country. This evaluation is performed in terms of the contribution that the designed wind farms would have to the total installed capacity in the country. This impact is calculated through the analysis of current installed capacities. Additionally, projected offshore wind farms are assumed to have a capacity factor similar to the average wind capacity factor for each country under analysis in order to verify how the energy generated from these could contribute to their total year demand.

## 4. ARCWIND Use Cases

The study of the impact on the grid connection was performed for all areas considered in ARCWIND. The project includes five target countries, covering all Atlantic areas of Europe in six different regions. High potential areas were identified in all targeted countries as presented in Table 1. These are translated directly into six use cases, which were analyzed by applying the methodology described in the last section.



### 4.1. Power System Models

The methodology implemented to calculate the grid impact on the point of connection of the designed wind farms required the existence of power system models, which is a requisite to perform the nodal capacity calculation process. These models need to allow the execution of steady state simulations, namely power flows, in order to check the existence of possible congestion in the system, which determine the nodal capacities.

As the targeted areas are from different countries and geographic areas, individual power system models had to be created or adapted for each use case individually. Table 2 contains a summary of power system models, highlighting their sources and means used to validate them.

**Table 2.** Power system models considered in ARCWIND.

| Country/Area | Source | Validation |
|---|---|---|
| Portugal | REN—internal TSO dataset | Verification of congestion |
| Spain—Ribadeo | ENTSO-E dataset [19] | Format conversion, verification of congestion |
| Spain—G. Canaria | In-house development | Model development and power flow validation using REE operational data [20,21] |
| France | ENTSO-E dataset [19] | Format conversion, verification of congestion |
| Ireland | In-house development | Model development and power flow validation using EirGrid operational data [22,23] |
| Scotland | In-house development | Model development and power flow validation using NG-ESO operational data [24,25] |

As can be seen in Table 2, used power system models have different sources. This also represents a different validation effort in order to ensure that each model has the consistency required to perform this study.

The power system model used for Portugal was directly provided by REN, the Portuguese TSO. This model is used by REN in their internal studies and therefore was already validated. The only required activity was the execution of preliminary simulations in order to verify that the system did not include any congestion, which is a requisite of the nodal capacity calculation tool.

The power systems for Spain (continental) and France were obtained from ENTSO-E, the European Network of Transmission System Operators for Electricity, which uses models provided by the different TSOs that compose this association. This model is available through the signature of an NDA which allows its utilization for R&D purposes without disclosing the model itself [19]. These models are also officially used by REE and RTE, the Spanish and French TSO, respectively. Because the model is provided by ENTSO-E in CGMES format, a conversion was performed in order to obtain a similar model in PSSE format, which is the required one for the nodal capacity tool. In addition to this conversion, and similarly to the Portuguese case, a few changes were made in the existing power flows in order to eliminate existing congestion. These included the increase in rated power of transformers far away from the area under study and do not have any measurable impact on the outcomes of the performed simulations. These three models are realistic models used by the TSOs and include detailed representations of the corresponding power systems, with realistic parametrization of its components (e.g., lines, transformers, and load levels).

The models for Gran Canaria, Scotland, and Ireland were completely developed in the scope of this activity. This need derived from the fact that there are no public models for these power systems. The models were created taking into account public information provided by the TSOs of these countries, namely REE, NG-ESO, and EirGrid. The creation of these models included two main activities:

- Modeling realistic grid topologies;
- Modeling operational (load and generation) scenarios.

Grid topologies were modeled taking as input the public network maps for transmission systems of these areas. As one example, Figure 1 depicts the map of the Gran Canaria

transmission network which was replicated in the created model. As there is no public information available, line lengths were calculated using the physical distance between the substations and average parameters were used for their characterization (e.g., line resistance and reactance). These average parameters were obtained from realistic characteristics of overhead lines for the considered voltage levels. Additionally, these maps were also used to locate main generation units, as can also be seen in Figure 1.

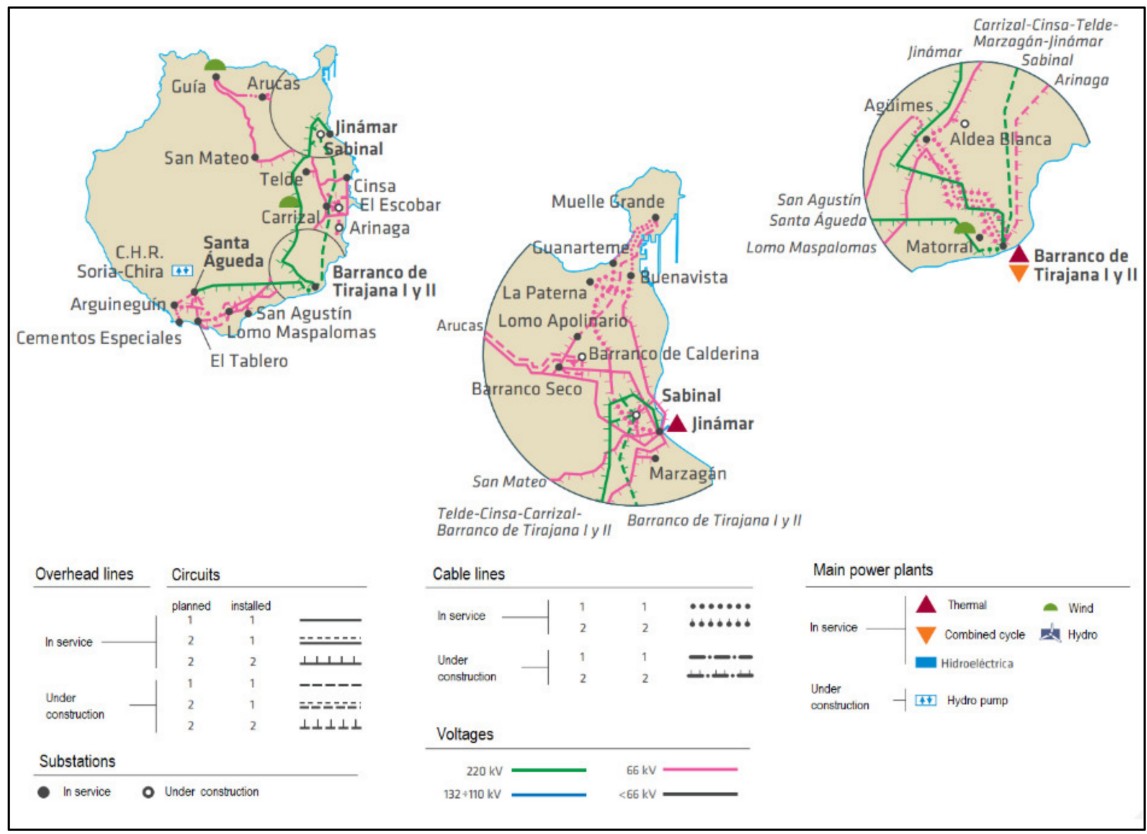

**Figure 1.** Gran Canaria transmission network map. Source: [26].

After the creation of the grid topologies, other technical parameters were obtained from the operational data of the TSOs in order to create realistic operational scenarios to be considered by the tool. These consider the load and generation levels. Since the location of most important generators is already included in the grid maps (see example of Figure 1), the generation values were included according to each generation unit main characteristics. Complementarily, load levels were split among the different substations either using the data provided by the TSOs (when there are available load levels per area) or using a direct correlation with urban density, increasing load in urban areas. A final validation of these models included the simulation of power flows in different operational conditions (e.g., using contingency analysis) in order to assess their robustness and accuracy when compared to operational data available from the TSOs. Figure 2 depicts an example of a power flow performed in Gran Canaria, where one can see the power flow and voltage profile of a small portion of the network.

The considered power systems are heterogeneous, being different in their size, energetic mix, and load levels, which is a natural consequence of the countries' differences. In order to allow the analysis performed in this study, Table 3 includes a high-level description of the characteristics of each one. Values included herein reflect the situation as of 31 December 2020 and were collected from TSO public databases.

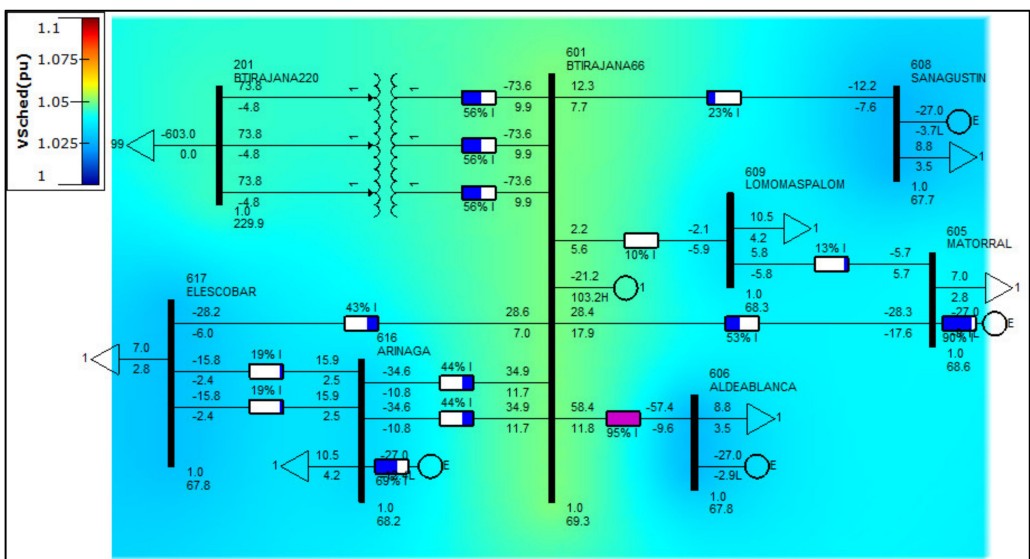

**Figure 2.** Power flow analysis in the Gran Canaria network.

**Table 3.** Power system models considered in ARCWIND.

| Country/Area | Total Installed Capacity (GW) | Wind Installed Capacity (GW) | Total Load Supplied in 2020 (TWh) |
|---|---|---|---|
| Portugal | 20.4 | 5.2 (0 offshore) | 48.8 |
| Spain—Continental | 105.6 | 27 (0 offshore) | 249.99 |
| Spain—G. Canaria | 1.26 | 0.194 (0 offshore) | 3.34 |
| France | 132.2 | 16.6 (14 MW offshore) | 460 |
| Ireland | 12.4 | 4.25 (0 offshore) | 30 |
| Scotland | 12.2 | 9.35 (0.98 GW offshore) | 30.3 |

*4.2. Candidate Grid Nodes*

The selection of a set of candidate nodes for each location followed the principles described in Section 3.1. As an example of this selection process, the methodology is explained herein for the selection of candidate nodes in the case of Spain (Ribadeo). Other use cases followed a similar methodology and candidate nodes for all cases are included in Table 4.

**Table 4.** Candidate nodes including substation name and voltage levels for all use cases.

| Country/Area | Substations | Voltage Level |
|---|---|---|
| Portugal | Estarreja (220 kV), Lavos, Paraimo | 400 kV |
| Spain—Continental | Boimente, El Palo, Pesoz, Xove | 400 kV |
| Spain—G. Canaria | Barranco de Tirajana | 220 kV |
| France | La Martyre, Pleyber Christ, Plaine-Haute, Rospez, Tregueux | 220 kV |
| Ireland | Tullabrack, Moneypoint, Slievecallan | 110 kV |
| Scotland | Cassley, Dounreay | 132 kV |

Figure 3 depicts the electricity transmission network near the high potential area for floating offshore wind farm deployment in Ribadeo, Spain, with this area highlighted as well. A high level model of the network topologies for this and other areas is available in [27].

The process of selection of candidate nodes for this location followed the next steps. Considering the targeted installed capacity of 880 MW, only 400 kV substations were considered as candidates. This is also in line with the available network assets in this area (where red lines correspond to 400 kV voltage level and green to 220 kV voltage level).

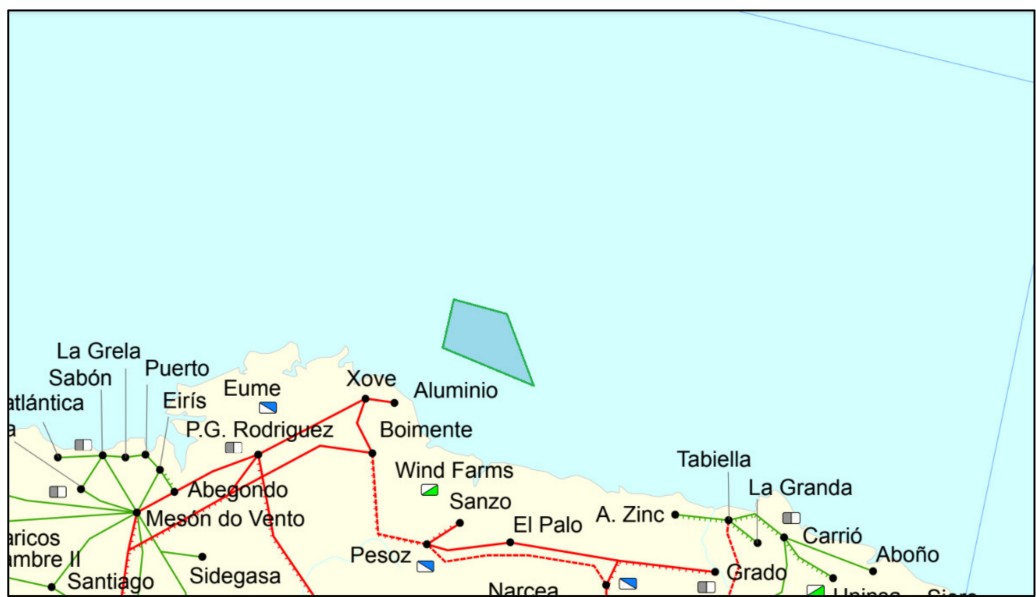

**Figure 3.** Spanish electricity transmission network near Ribadeo.

Based on this analysis, seven potential candidate nodes were identified: the four included in Table 4 and P.G. Rodriguez, Aluminio, and Sanzo. However, P.G. Rodriguez and Aluminio were excluded because there is a major generation facility and an industrial load facility connected to these substations, respectively. Sanzo was excluded as this is the main connection point of several onshore wind farms that already exist in the area. Thus, the application of these criteria, which was described in Section 3.1, resulted in the candidate nodes included in Table 4.

## 5. Results

The calculation of the nodal capacity in selected grid nodes as well as a connection impact assessment was performed for the six use cases considered in ARCWIND.

The nodal capacity calculation process, described in Section 3.1, was applied to the different sets of candidate nodes in order to identify the best point of connection (the best candidate) for projected wind farms in each one of the identified high potential areas. Table 5 includes the results of this analysis for all candidate nodes in the six use cases.

The best candidates were selected based on the available nodal capacity and the geographic location to the high potential area. These are indicated in bold in Table 5.

In the cases of Portugal, France, Ireland, and Scotland, the nodes identified as best candidates are the ones with a higher nodal capacity. In this context, the nodes considered as best candidates are proposed as the selected grid nodes for the connection of the offshore wind farms, in the case of a future connection and are the ones selected for the connection impact assessment.

In Spain (Ribadeo), the substation of Boimente was selected because it had a large capacity (almost two times the targeted installed one in the offshore area) and was closer to this area than the Pesoz substation which would be the other potential best candidate. In the case of Gran Canaria, a single node was considered as a candidate due to its location being close to the identified high potential area and including an adequate voltage level. Obtained values indicate that the Barranco de Tirajana substation has sufficient nodal capacity for the connection of the possible offshore wind farm.

In order to demonstrate the accuracy and applicability of the nodal calculation process, Figure 4 depicts the results obtained for the simulations performed in the Spanish transmission system, near the Ribadeo area. Nodal injections are highlighted in green and it can be seen that there are multiple lines with power flows near their maximum rated capacity, which acts as the limiting factor for the nodal capacity calculation process.

**Table 5.** Nodal capacity calculation results.

| Country/Area | Candidate Node | Nodal Capacity (MW) |
|---|---|---|
| Portugal | Estarreja (220 kV) | 129 |
| | **Lavos (400 kV)** | **139** |
| | Paraimo (400 kV) | 112 |
| Spain—Continental | **Boimente (400 kV)** | **1505** |
| | El Palo (400 kV) | 844 |
| | Pesoz (400 kV) | 1846 |
| | Xove (400 kV) | 781 |
| Spain—G. Canaria | **Barranco de Tirajana (220 kV)** | **603** |
| France | La Martyre (220 kV) | 587 |
| | Pleyber Christ (220 kV) | 571 |
| | Plaine-Haute (220 kV) | 243 |
| | **Rospez (220 kV)** | **776** |
| | Tregueux (220 kV) | 462 |
| Ireland | Tullabrack (110 kV) | 205 |
| | **Moneypoint (110 kV)** | **380** |
| | Slievecallan (110 kV) | 219 |
| Scotland | Cassley (132 kV) | 235 |
| | **Dounreay (132 kV)** | **316** |

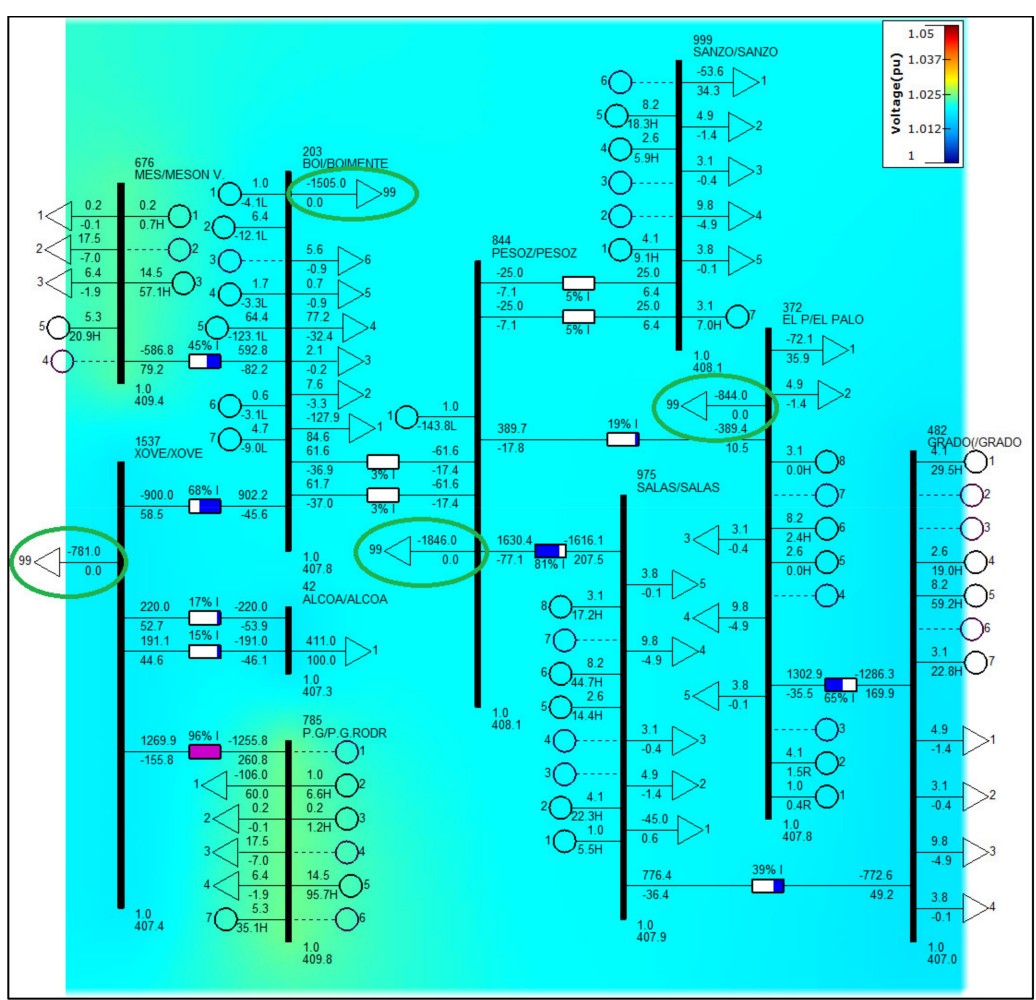

**Figure 4.** Nodal capacity calculation results for Spain (Ribadeo area).

After the identification of the best candidate nodes, the analysis of the impact of the connection in these nodes was performed. This set of simulations, particularly focused in simulating short circuits, considers that an equivalent generator is connected to the selected grid node, with a power generation corresponding to the wind farm maximum capacity, as presented in Table 1. In order to measure the impact of the connection, three-phase short circuits were simulated in the point of connection and neighbor substations in two different conditions: without the connection of the wind farm (base case) and with the wind farm connected. This allows to measure the contribution of the wind farm to short circuit current levels, which can dictate the need to refurbish protection system components as the circuit breakers. The neighbor substations do not always match the previously selected candidate nodes exactly. This is related to the grid topologies.

The outcome of the calculation of short circuit current (SCC) levels is presented in Table 6. This analysis does not include the cases of Scotland and Ireland as the created grid models (which are based on average parameters and open-source data) do not allow for the necessary parametrization and short circuit values obtained would not have any physical meaning.

**Table 6.** Short circuit current analysis.

| Country/Area | Substation | Base Case SCC (kA) | Wind Farm Case SCC (kA) | Delta |
|---|---|---|---|---|
| Portugal | Lavos | 16.44 | 18.25 | 9% |
| | Batalha | 15.27 | 16.78 | 9.1% |
| | Paraimo | 17.98 | 19.81 | 9.1% |
| Spain—Continental | Boimente | 22.54 | 23.27 | 3.2% |
| | El Palo | 18.61 | 18.76 | 0.8% |
| | Pesoz | 21.51 | 21.75 | 1.1% |
| | Salas | 21.80 | 21.92 | 0.6% |
| | Sanzo | 21.16 | 21.40 | 1.1% |
| | Xove | 20.15 | 20.58 | 2.1% |
| Spain—G. Canaria | Barranco de Tirajana | 4.42 | 5.23 | 15.8% |
| France | Rospez | 9.61 | 9.65 | 0.4% |
| | Plaine-Haute | 23.21 | 23.36 | 0.6% |
| | Tregueux | 8.15 | 8.20 | 0.9% |
| | Brennilis | 12.08 | 12.42 | 0.8% |

Results obtained for Spain (Ribadeo) and France indicate that there are marginal increases in short circuit current levels, which should not have any impact on the existing protection systems. In the case of Gran Canaria, the increase of 15.8% in the short circuit current levels could result in the need to refurbish the circuit breakers. In the case of Portugal, the connection of the wind farm would result in a maximum increase of 9.1% which will be further discussed in Section 6.

The last activity performed within the scope of this work aimed to verify the impact on the energy mix of each country as well as possible contributions to renewable energy generation to fulfill yearly demand values. This analysis assumes that the floating offshore wind farms would have similar capacity factors to the current ones at national level, which can be considered as a conservative approach (as most installed wind power is onshore). Table 7 contains the results of this analysis, for the five different countries. As can be seen, the projects wind farms would have significant impacts on the different countries, supplying electric load with non-negligible values, having a significant impact on the increase in renewable energy generation in the five countries and consequent reduction in energy generated from fossil fuels.

**Table 7.** Impact on energy mix.

| Country/Area | Impact on Current Wind Installed Capacity (%) | Annual Energy Generation (TWh) | % of Load Supplied (TWh) |
|---|---|---|---|
| Portugal | 13.5% | 1.7 | 3.5% |
| Spain—Continental | 3.3% | 2.2 | 0.9% |
| Spain—G. Canaria | 61.9% | 0.3 | 9.0% |
| France | 2.8% | 1.0 | 0.2% |
| Ireland | 2.3% | 0.3 | 0.8% |
| Scotland | 1.6% | 0.4 | 1.2% |

## 6. Discussion

The obtained results seem to indicate that the offshore wind farms designed in AR-CWIND could contribute actively to the generation of wind energy; thus, contributing to the ongoing decarbonization process in the five studied countries.

In general, performed studies indicate that it seems to be possible to integrate the designed wind farms into existing power systems, with no grid reinforcement needs in the case of Spain (Ribadeo), France, Scotland, and Ireland. In the case of Portugal, the integration would only be possible with significant grid expansion measures. Since Lavos was considered as the best candidate node and selected for connection impact assessment, preliminary results obtained using the set of operational scenarios considered in this study indicate the need to reinforce the lines between Lavos and Batalha and Paraimo substations. In the case of Gran Canaria, although the nodal capacity calculation process indicates that the wind farm could be connected without the need for grid reinforcements, there is some operational risk in this connection. The installed capacity (120 MW) corresponds to 54% of the minimum load and 22% of peak load verified in 2020. This represents possible curtailment needs for the wind farm to retain the operational security of the system.

In order to further demonstrate the validity of the implemented methodology and obtained results, a comparison with official data from REN, the Portuguese TSO, is presented. This was performed for the nodal capacity and short circuit current calculation methods.

REN biannually publishes the current capacity of the network. These are included in the network development plan, whose last version, from 2021 and reflecting the period of 2022–2031, is available in [4]. The methodology used from REN is slightly different from the one presented in this document. REN publishes the nodal capacities using a zonal approach, i.e., for a set of geographically close nodes and not for individual ones. Since no public information is available for individual nodes, it can be assumed that the individual capacities for each node are the average value for that zone, being added to obtain the total zonal value.

The nodes considered as candidate nodes in the study performed in ARCWIND belong to two different zones. The zonal capacity of these two zones is presented in Table 8. This table also includes the individual nodal capacities calculated in this study.

**Table 8.** Nodal capacity calculation in Portugal: comparison with REN data.

| Zone | Node | REN Zonal Capacity (MW) | ARCWIND Capacity (MW) |
|---|---|---|---|
| Zone A | Paraimo | 340 (113.3 average per node) | 112 |
| | Estarreja | | 119 |
| | Mourisca | | N/A |
| Zone B | Lavos | 696 (174 average per node) | 139 |
| | Pombal | | N/A |
| | Batalha | | N/A |
| | Rio Maior | | N/A |

Comparing the values calculated in ARCWIND with those given by REN, it can be concluded that the methodology used in this work provides valid technical results. The differences obtained for the different nodes (1.1% in Paraimo, 13.8% in Estarreja, and 20.1% in Lavos) result from the fact that this study only considers a small subset of four operational scenarios (generation and load conditions) while REN uses dozens of different scenarios in their studies.

In addition to the zonal capacities, REN also includes in their network expansion plans the maximum and minimum short circuit currents calculated for the considered operational scenarios. These are given in Table 9, together with the previously calculated values in this study. As can be seen, both SCC levels for the base case and considering the connection of the wind farm are within the interval given by REN, which demonstrates the applicability of the methodology. Since these are within the interval defined by REN, even if there is an increase of 9.1% in short circuit currents with the connection of the wind farm, there is no need for refurbishment of protection systems.

**Table 9.** Short circuit current calculation in Portugal: comparison with REN data.

| Substation | Base Case SCC (kA) | Wind Farm Case SCC (kA) | REN SCC Interval (kA) |
|---|---|---|---|
| Lavos | 16.44 | 18.25 | (15.7–22.8) |
| Batalha | 15.27 | 16.78 | (14.8–19.8) |
| Paraimo | 17.98 | 19.81 | (16.8–20.9) |

## 7. Conclusions

This work presented a study related to the impact assessment at the grid connection point for different sized floating offshore wind farms considered in the scope of the ARCWIND project. This study included two main activities related to the calculation of the capacity that the networks have to absorb the power generated by those wind farms and the impact that they would have on the system, both at local and national level.

Results obtained seem to indicate that, with the exception of the Portuguese Case, all other considered networks have enough capacity to perform the connection of the wind farms without the need for major grid reinforcements. Looking into possible operation constraints, the only case that could result in curtailment needs is the connection of the wind farm in Gran Canaria Island, due to the considered wind farm size when compared to the operational conditions of the power system (load levels). Additionally, the calculation of short circuit currents also shows a possible need to refurbish part of the existing protection devices in Gran Canaria. Other cases would not need any change in these systems.

The installation of these floating wind farms, given that the necessary environmental and economic viability studies were positive, would have an important technical contribution to the ongoing energy transition process, by substantially increasing the installed wind capacity and generation levels in the five countries.

**Author Contributions:** N.A. contributed with the development of the methodology, writing of the paper, and analysis of obtained results. A.E. contributed with the creation of some of the power system models and the execution of the simulations for the nodal capacity calculation process. G.G. performed the short circuit simulation and analysis. All authors have read and agreed to the published version of the manuscript.

**Funding:** This research was partially funded by INTERREG Atlantic Area through the project ARCWIND (EAPA 344/2016).

**Institutional Review Board Statement:** Not applicable.

**Informed Consent Statement:** Not applicable.

**Conflicts of Interest:** The authors declare no conflict of interest.

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
