# Peer review of "Fostering Offshore Wind Integration in Europe through Grid Connection Impact Assessment"

_jmse, doi:10.3390/jmse10040463_

Round 1

Reviewer 1 Report

jmse-1627545_reviewer

In this article, the authors discuss about Floating offshore wind energy is one of the solutions which can foster the ongoing climate transition in Europe, The manuscript is written very well. The presented information’s are based on references that are updated; the work is well-structured, well-written and easy to understand. It also addresses a subject that is of great interest in the scientific community

my remarks are:

1- review figure 1, 2 and 4 are not visible.

3- add the date of consultation of this reference.  14. EDP, “WindFloat Project,” 2018. https://www.edp.com/en/windfloat

4-some remarks to reference frome 22 to end.

and i have some questions

1- Without taking care of the calculations made in your opinion, how much you find that the Marine Floating energy meets the energy need.
2- can a marine flotation installation be finer if compared to an onshore installation?

Therefore I recommend publishing this paper in our journal after all remarks are add.

Author Response

Thank you or your revision.

Regarding your remark 1, we have improved the quality of figures 1, 2 and 4.

Regarding remark 3 and 4, we have included the date of access for all websites.

Please find attached a new version of the paper with all comments addressed.

As for your particular questions, please find some (as short as possible) answers:

1 - We believe that Marine energies (not only wind) can be an important contribution to meeting the decarbonisation targets. floating wind will be one of these main vectors in countries that do not allow for fixed platforms. Typical capacity factors are higher than those of onshore installations and with the R&D developments in building larger turbines (even higher than 10MW each), future floating wind farms can be a main contributor to the energy mix in many countries.

2 - as mentioned in 1), capacity factors for offshore wind are usually higher than those of onshore. Additionally, in some countries onshore have already reached almost maximum installation capacities (due to landscape constrains). this, together with the fact that usually offshore wind considers larger turbines, can demonstrate the applicability of floating offshore wind farms. Currently they have a larger cost that the onshore one, but with their possible increased installed capacity, they'll become more and more competitive.

Reviewer 2 Report

This article includes the description of offshore wind integration as well as its application to six different use cases covering five European countries: Portugal, Spain, France, United Kingdom and Ireland. The work is interesting, and the structure is clear. This paper could be accepted.

Author Response

Thank you for your revision. AS there were no specific comments, we just included some minor updates to the paper text (correcting some typos and grammar problems).